# Genome-Wide Analyses of *CCHC* Family Genes and Their Expression Profiles under Drought Stress in Rose (*Rosa chinensis*)

**DOI:** 10.3390/ijms25168983

**Published:** 2024-08-18

**Authors:** Shijie Li, Jun Xu, Yong Cao, Jie Wu, Qing Liu, Deqiang Zhang

**Affiliations:** 1School of Landscape Architecture, Beijing University of Agriculture, Beinong Road 7, Huilongguan, Changping District, Beijing 102206, China; aaalishijie@163.com (S.L.); junxu0101@163.com (J.X.); 15953222998@163.com (Y.C.); wjwj1106@163.com (J.W.); 2CSIRO Agriculture and Food, Black Mountain, Canberra, ACT 2601, Australia; qliu2102002@gmail.com

**Keywords:** zinc finger proteins, cut flowers, abiotic stress, VIGS, gene family analysis

## Abstract

CCHC-type zinc finger proteins (CCHC-ZFPs), ubiquitous across plant species, are integral to their growth, development, hormonal regulation, and stress adaptation. Roses (*Rosa* sp.), as one of the most significant and extensively cultivated ornamentals, account for more than 30% of the global cut-flower market. Despite its significance, the *CCHC* gene family in roses (*Rosa* sp.) remains unexplored. This investigation identified and categorized 41 *CCHC* gene members located on seven chromosomes of rose into 14 subfamilies through motif distribution and phylogenetic analyses involving ten additional plant species, including *Ginkgo biloba*, *Ostreococcus lucimarinus*, *Arabidopsis thaliana*, and others. This study revealed that dispersed duplication likely plays a crucial role in the diversification of the *CCHC* genes, with the Ka/Ks ratio suggesting a history of strong purifying selection. Promoter analysis highlighted a rich presence of cis-acting regulatory elements linked to both abiotic and biotic stress responses. Differential expression analysis under drought conditions grouped the 41 *CCHC* gene members into five distinct clusters, with those in group 4 exhibiting pronounced regulation in roots and leaves under severe drought. Furthermore, virus-induced gene silencing (VIGS) of the *RcCCHC25* member from group 4 compromised drought resilience in rose foliage. This comprehensive analysis lays the groundwork for further investigations into the functional dynamics of the *CCHC* gene family in rose physiology and stress responses.

## 1. Introduction

Roses (*Rosa* sp.) constitute a significant segment of the global ornamental flower market, accounting for more than 30% of total cut-flower sales worldwide. Notably, *Rosa chinensis* Old Blush, first introduced from China to Europe in the 18th century, has played a pivotal role in the history of rose breeding [1]. Today, *R. chinensis* remains a vital model plant for rose research, continuing to provide essential insights into the genetics and breeding of roses. Recent research has highlighted that abiotic stress significantly affects the growth and ornamental value of roses. Under drought stress, the length of rose (*Rosa* spp.) branches is curtailed, and the leaves manifest wrinkling, reduced thickness, and size. Initially, the chlorophyll content and superoxide dismutase (SOD) activity decline, but subsequently exhibit resurgence. Concurrently, the plant’s reactive oxygen species (ROS) metabolism is suppressed, and the catalase (CAT) activity exhibits erratic changes [2]. Current research on the core transcription factors that respond to rose drought stress includes WRKY, MYB, NAC, ERF, ARF, and bHLH. Specifically, *RhNAC3* enhances petal desiccation resistance via osmotic gene regulation and abscisic acid (ABA) signaling, while *RhABF2/RhFer1* improves tolerance by sequestering free iron ions under stress [3,4]. Nonetheless, research on the zinc finger proteins (ZFPs) family, which is crucial in stress response, has been little studied in roses.

In higher plants, zinc finger proteins (ZFPs) are one of the most common transcription factors (TFs). The defining characteristic of these proteins, known as the zinc finger (ZF) domain, involves a zinc ion coordinated by cysteine (C) and histidine (H) residues [5]. ZFPs are divided into numerous varieties depending on the arrangement and number of these coordinating residues, including C2H2, C2HC, C2C2, C2HC5, C3HC4, CCCH, C4, C6, and C8 [6]. Among the earliest identified ZFPs was TFIIIA in *Xenopus laevis*, which is a ubiquitous TF characterized by nine C2H2 motifs, capable of binding both DNA and RNA [7]. Subsequent discoveries across various plants have demonstrated the critical role of ZFPs in stress response, reproductive development, and vegetative growth [8,9].

CCHC-type zinc finger proteins (CCHC-ZFPs) form a crucial subset within the diverse ZFP family, prevalent across plants, animals, and microorganisms. These proteins are characterized by at least one zinc knuckle motif, typically conforming to the consensus sequence CX2CX4HX4C, where ‘X’ stands for any amino acid [10]. This specific motif highlights the nuanced variations within the ZFP family, illustrating how different structural configurations are tailored to distinct biological functions. Understanding these variations enables researchers to grasp the breadth and specificity with which ZFPs operate within biological systems. For example, the RNA recognition motif (RRM) domain-containing *PtoRSZ21* gene plays a pivotal role in the plant’s resistance to drought stress by positively regulating drought tolerance in Populus by controlling the alternative splicing of *PtoAT2b* and modulating stomatal size [11], while the seed size of *Medicago truncatula* is regulated by the *Mt-Zn-CCHC* gene [12]. Up to now, the regulation of RNA metabolism has been linked to the ZCCHC family in an increasing number of studies [13]. In the study of the mechanisms of drought stress tolerance in barley (*Hordeum vulgare* L.), ZFPs (zinc finger proteins) are particularly noted as a class of transcription factors with a key role. The upregulation of C2H2 zinc finger transcription factors under drought stress implies their potential involvement in regulating the plant’s adaptive response to aridity, activating or suppressing a series of downstream genes, thereby aiding the plant in acclimating to arid conditions [14]. In the same species, overexpression of *AtCSP4* leads to reduced embryo lethality and silique length [15]. Similarly, abiotic stresses, including exposure to salt (NaCl), alkali (NaHCO_3_), and hydrogen peroxide (H_2_O_2_) treatment, elevate the expression of *OsZFP6* [16]. Furthermore, the ectopic expression of wheat *RaRZ1* in *Arabidopsis thaliana* increases resistance to bacterial invasion, indicating *CCHC-ZFP*’s role in biotic stress resilience [17]. Additionally, the study found that genes related to ABA and auxin synthesis and signal transduction also underwent significant changes after drought stress treatment. Under drought stress, ABA can regulate the levels of enzymes that scavenge reactive oxygen species (ROS), while strigolactones (SLs) affect stomatal closure by modulating plant sensitivity to ABA, manipulate chlorophyll components, and activate antioxidant capacity to reduce the negative impacts of drought [18]. Despite the recognized importance of the *CCHC* gene family in plant genetics, comprehensive and systematic research has been limited, with notable reports only from yeast, humans, *Ustilaginoidea virens*, *A. thaliana*, and wheat [19,20,21]. Up to now, no studies have explored the *CCHC* gene family in roses, presenting a notable gap in the research.

Therefore, there is an impetus to investigate the resistance genes in roses to enhance breeding and production strategies. In light of the crucial function of *CCHC-ZFP* genes, an in-depth examination of the *R. chinensis CCHC-ZFP* (*RcCCHC-ZFP*) gene family could provide vital insights into enhancing stress resistance in roses. This study aims to delineate the *RcCCHC-ZFP* gene family through genomic bioinformatics and to analyze its various aspects. These efforts set the platform for future research on the *RcCCHC-ZFP* genes, offering a theoretical basis for developing stress-resilient rose varieties.

## 2. Results

### 2.1. Identification and Analysis of CCHC Genes

In this study, we identified a total of 41 members of the *RcCCHC-ZFP* gene family in the rose genome (Figure 1). Comprehensive data including subcellular localization, molecular weight (MW), instability index, and grand average of hydropathicity (GRAVY) are presented in Table 1. The CCHC family encodes proteins with an average length of 399 amino acids, ranging from 164 amino acids (*RcCCHC2*) to 1762 amino acids (*RcCCHC18*). The instability index spanned from 25.94 (*RcCCHC2*) to 108.65 (*RcCCHC7*). Notably, the majority of the GRAVY values for RcCCHC proteins were negative, indicating their hydrophilic nature. Subcellular localization prediction indicated that 19, 2, 3, 15, and 2 RcCCHC-ZFPs localized to the nucleus, mitochondria, Golgi apparatus, chloroplasts, and cell wall, respectively. Additionally, using HMMER3.2.2 software, we searched for *CCHC* genes in ten other species: *Ginkgo biloba*, *Ostreococcus lucimarinus*, *A. thaliana*, *Oryza sativa* subsp. *japonica*, *Triticum aestivum*, *Solanum lycopersicum*, *Vitis vinifera*, *Fragaria ananassa*, *Prunus persica*, and *Rosa rugosa*. The findings confirm a significant variation in the distribution of *CCHC* genes across Plantae, with *R. rugosa* and *P. persica* exhibiting the highest numbers, 88 and 89, respectively, while *O. lucimarinus* displayed the fewest (Appendix A).

### 2.2. Gene Structure and Conserved Motifs Analysis

The 41 *RcCCHC-ZFP* genes were categorized into 14 distinct subfamilies according to the phylogenetic analysis (Figure 2A). Using the WebLogo online tool, we generated a sequence logo for the CCHC domain (Appendix A), revealing similarities to Arabidopsis and wheat yet distinct differences from yeast and humans [19]. Notably, glycine (G) predominates at the tenth position within the conserved CCHC domain.

Employing the MEME suite, we identified ten conserved motifs across the *RcCCHC-ZFP* genes (Figure 2C). Of these, motifs 1, 4, and 7 correspond to the CCHC domain, motif 5 to the RNA recognition motif (RRM), and motif 6 to the retrotransposon-gag domain. The distribution of motifs varied significantly, with 1 to 7 motifs present per gene, depending on the subfamily. Subfamilies XIII, XIV, and XV exhibit a higher prevalence of the CCHC domain (5–7 motifs), whereas other subfamilies typically display 1–2. Additionally, motifs 2, 3, and 9 are exclusive to subfamily VI, motif 10 to subfamily X, and motif 5 predominantly to subfamily XVII, suggesting conserved motif patterns within the subfamilies.

Further investigation into the exon-intron structures of *RcCCHC-ZFP* genes revealed exon counts ranging from 1 to 14 (Figure 2B). Furthermore, it is notable that genes belonging to the same subfamily tend to exhibit structural similarities. For instance, all genes in subfamily VI have two exons whereas those in subfamily XVI possess only one. Subfamilies I, XI, XII, XIII, XIV, and XVII, display more complex gene structures. This structural diversity among *RcCCHC-ZFP* genes likely reflects their diverse functional variability (Appendix A).

### 2.3. Chromosomal Location and Gene Duplication Analysis

The 41 identified *RcCCHC-ZFP* genes are unevenly distributed across the seven chromosomes of rose (Figure 3). The genes predominantly localize to the chromosome extremities, forming nine gene clusters. Chromosome 1 exhibits the densest distribution with 13 genes, followed by chromosome 7, which contains 7 genes. In contrast, chromosomes 2 and 5 host a minimal number of *RcCCHC-ZFP* genes, each harboring only three.

Gene duplication serves as a crucial mechanism in plant evolution, offering a rich source of material for gene family expansion. Various processes can lead to gene duplication, including whole-genome duplication (WGD), proximal duplication (PD), dispersed duplication (DSD), retrotransposed duplication (TRD), and tandem duplication (TD). We employed the DupGen_finder pipeline to analyze these five modes of duplication in the evolutionary trajectory of *RcCCHC-ZFP* genes (Appendix A). Our analysis identified only 25 gene pairs (0.08%) exhibiting duplication within the rose *CCHC* gene family, highlighting its high conservation. Of the duplication types, DSD accounts for the majority with 18 pairs, whereas PD is represented by just a single gene pair on chromosome 7. Notably, no instances of TD were observed among the *RcCCHC-ZFP* genes. Collectively, these findings suggest that the rese *CCHC* gene family is primarily driven by DSD for expansion.

### 2.4. Phylogenetic Analysis

A phylogenetic tree based on the maximum-likelihood (ML) method was constructed using 450 CCHC-ZFP sequences from various species, categorized into 17 subfamilies (I to XVII) (Appendix A, Appendix A). Meanwhile, a small ML phylogenetic tree was constructed using *CCHC-ZFP* genes from *Rosa chinensis*, *Rosa rugosa*, and *Arabidopsis thaliana*, with subfamilies of each gene marked on the outer circle (Figure 4). Similarly, the distribution of CCHC-ZFPs across these subfamilies is notably uneven, with subfamilies XVII and X containing the most sequences, 99 and 80 genes, respectively, and subfamilies IV the fewest, with only 7 in all species.

In the unicellular alga *O. lucimarinus*, four *CCHC-ZFP* genes were identified, suggesting that the emergence of these genes precedes the algal diversification. Excluding algae, the prevalence of *CCHC-ZFP* genes remains relatively consistent across plant species, indicating a high degree of evolutionary conservation. Current research posits that subfamilies X and XVI might be involved in the synthesis and metabolism of plant hormones [22,23], whereas subfamily XI is implicated in plant stress tolerance [16]. The results of two phylogenetic trees demonstrate that genes with the same domain often cluster together stably and perform similar functions (Appendix A). For example, subfamily XVII contains the RRM domain, associated with mRNA post-transcriptional processing [24], and subfamily XVI includes the cold-shock domain (CSD), linked to plant responses to low-temperature stress. Subfamily X contains the Retrotrans-gag domain, suggesting involvement in reverse transcription process [25].

Significantly, subfamily VII is exclusive to peach, showing significant duplication. In a lineage-specific amplification within the *CCHC* family, substantial numbers of genes were identified: 45 in *R. rugosa* and 20 in peach genes within subfamily X, 28 in wheat within subfamily XVII, and 16 in *R. rugosa* within subfamily V. This indicates that the functional attributes of wheat, peach, and *R. rugosa* are significantly influenced by these genes.

### 2.5. Collinearity and Evolution of CCHC Gene Family in Rose and Other Species

In order to gain further insight into the evolutionary dynamics and genesis of the *RcCCHC-ZFP* genes, we conducted a comparative analysis involving 41 rose *CCHC* genes and 10 other species, encompassing 1 alga, 1 gymnosperm, 2 monocots, 3 dicots, and 3 Rosacase plants (Appendix A, Appendix A). Collinearity analysis between rose and four representative species revealed 25 genes with collinearity. Conversely, 16 genes appeared to be unique to roses among the analyzed *CCHC* genes. Notably, 23 homologous gene pairs were identified in strawberry and 16 in Arabidopsis, while rice and ginkgo exhibited only 2 and 4 homologous pairs, respectively (Figure 5A). There were no homologous gene pairs identified in wheat and *O. lucimarinus*. Consistent with the phylogenetic findings, these results highlight the pronounced divergence between monocots and dicots within the *CCHC* gene family, reflecting selective pressures that have steered their evolution in disparate directions.

Further insights were gained by calculating the Ka/Ks ratio of these homologous gene pairs using TBtools v2.11.2 (Appendix A). Generally, a Ka/Ks ratio less than 1 indicates purifying selection, a ratio greater than 1 suggests positive selection, and a ratio equal to 1 implies neutral selection [26]. Our analysis revealed that all homologous gene pairs had Ka/Ks ratios below 1, with the majority falling below 0.3, underscoring the significant functional role of *CCHC* genes in plant biology. These results imply that over the evolutionary history, the *RcCCHC-ZFP* genes most likely have undergone purifying selection to elimination harmful mutations. Additionally, the divergence times of these gene pairs align closely with those inferred from the species evolutionary tree, further elucidating the variability in gene pair numbers between rose and other species (Figure 5B, Appendix A).

### 2.6. Cis-Active Elements Analysis of RcCCHC-ZFP Genes

To further examine the regulatory mechanisms potentially governing *RcCCHC-ZFP* genes, we analyzed the promoters of all *RcCCHC-ZFP* genes using the PlantCARE online tool, identifying various *cis*-active elements (Appendix A). These elements were categorized into three primary groups: abiotic and biotic stresses, phytohormone responsive, and plant growth and development (Figure 6). Among the 41 evaluated *CCHC* genes, *RcCCHC25*, *RcCCHC33*, and *RcCCHC30* exhibited the highest number of cis-active elements, containing 58, 56, and 49 elements, respectively. A notable observation from the analysis is the predominance of MYB binding sites, which total 286 across the gene family. This suggests a significant interaction between the *CCHC* and *MYB* gene families in rose, potentially indicating a shared or complementary role in regulatory processes.

In our comprehensive examination of the 41 *RcCCHC-ZFP* genes, a total of 471 cis-active elements associated with phytohormone responsiveness and abiotic stress responses were predicted. Specifically, five types of hormone-responsive cis-elements were identified, including 108 methyl jasmonate (MeJA)-responsive elements (CGTCA-motif and TGACG-motif), 78 abscisic acid responsive elements (ABREs), 61 salicylic acid responsive elements (TCA-element, P-box, and GARE-motif), 26 auxin responsive elements (TGA element and AuxRR-core), and 8 gibberellin responsive elements (TATC-box). Additionally, five types of abiotic stress-responsive *cis*-elements were discovered, comprising 107 anaerobic induction elements (ARE), 45 drought responsive elements (MBS), 22 low-temperature responsive elements (LTR), 12 defense and stress responsive elements (TC-rich repeats), and 4 wound responsive elements (WUN-motif). These findings suggest that *RcCCHC-ZFP* genes are likely integral to plant growth and resilience against abiotic stress, operating through diverse regulatory pathways. This diverse array of *cis*-elements highlights the complex interaction between environmental stress factors and hormonal signals in regulating gene expression, thereby influencing plant adaptive responses.

### 2.7. Heatmap Analysis of RcCCHC-ZFP Genes under Drought Stress

To further explore the functional dynamics of *RcCCHC-ZFP* genes in response to drought stress, we constructed a heatmap of 41 *CCHC* genes across varying degrees of drought stress (moderate to severe) and in distinct tissues (leaves and roots) [27] (Figure 7A). The analysis revealed a universal modulation of those genes by drought stress. Distinct expression patterns across different tissues under varying stress levels led to the categorization of the *CCHC* genes into five groups. In roots, genes in groups 3, 4, and 5 predominantly exhibited upregulation in response to drought. Notably, genes in group 1 were consistently downregulated in both tissue types under drought conditions. Conversely, the expression levels of genes in group 2 (roots) and group 5 (leaves) remained unchanged, highlighting their tissue-specific roles. In leaves, genes in groups 2 and 4 demonstrated significant upregulation under drought stress, with group 2 genes being more pronounced during moderate drought and group 4 during severe drought. Of particular interest is the expression profile of *RcCCHC25*, which mirrored this pattern in both leaves and roots, suggesting a correlation between gene expression and the escalating severity of drought conditions. In addition, the expression levels of six genes were validated in rose leaves during distinct drought periods using RT-qPCR (Figure 7B–G).

### 2.8. Protein Interaction Network of RcCCHC-ZFP Genes

Protein–protein interaction (PPI) is critical for predicting the biological functions of proteins, particularly among those with similar functions. Utilizing the STRING online database, we predicted the PPI network of the *CCHC* gene family in rose (Figure 8). The network comprised eleven proteins interconnected by 58 links, indicating substantial interrelatedness among these proteins. Notably, the RcCCHC25 protein was involved in 14 connections, highlighting its significant role in regulating plant growth and stress resistance.

### 2.9. Silencing of RcCCHC25 Reduces Tolerance to Drought Stress

To validate the function of *RcCCHC25* in drought tolerance, we employed virus-induced gene silencing (VIGS) on rose, subjected to 20% PEG6000 for 72 h (h) (Figure 9A). The results demonstrated that there was no discernible phenotypic change between TRV and TRV-RcCCHC25 plants before drought treatment. However, following 72 h of drought, TRV-RcCCHC25 plants exhibited a pronounced degree of leaf wilting and shedding in comparison to TRV. To further verify whether the *RcCCHC25* gene plays a role in the resistance of rose leaves to drought stress, we conducted a new drought experiment using rose leaves. We employed virus-induced gene silencing (VIGS) on rose leaves, placed in dry glass Petri dishes for a 24 h dehydration period, followed by a 3 h rehydration period (Figure 9B). Subsequent analysis revealed a reduction in *RcCCHC25* transcript levels to 60% of the control (TRV leaves) (Figure 9C). Initially, both TRV and TRV-RcCCHC25 plants exhibited similar phenotypes. However, post-treatment TRV-RcCCHC25 leaves exhibited more pronounced leaf curling and brittleness compared to controls, with a 30% greater relative wilting area (Figure 9E). Moreover, ion leakage in TRV-RcCCHC25 leaves increased by about one-third compared to TRV controls (Figure 9D), indicating the *RcCCHC25*-silencing decreases rose drought tolerance.

## 3. Discussion

The *CCHC* gene family is crucial for plant growth and development, hormonal responses, and stress tolerance, with extensive research conducted in Arabidopsis and wheat [19,21]. Roses, significant in the ornamental plant sector, are notably vulnerable to environmental stresses during cultivation. Up to now, the *CCHC* gene family in roses has not undergone a comprehensive analysis. This study employed bioinformatics methods to thoroughly investigate the *RcCCHC* family in *R. chinensis*, laying a foundation for more detailed research into the function of these genes.

The genetic architecture and evolutionary trajectories exhibit considerable heterogeneity across species, often resulting in varied gene complements within orthologous gene families. We identified a total of 41 *RcCCHC* genes, a count exceeding that in many other species but fewer than in pear (89), *R. rugosa* (88), and wheat (50) (Appendix A). This variation suggests that the *CCHC* gene family has expanded to varying extents during evolution across different species. Additionally, we analyzed the conserved CCHC motifs within these genes, confirming the consistency of conserved sites with previous studies [19]. Studies indicate that the CCHC domain serves as a typical nucleic acid recognition motif, with these types of zinc fingers demonstrating strong affinity for binding to single-stranded nucleic acids [28]. Predominantly, CCHC zinc finger proteins (CCHC-ZFPs) are recognized as RNA-binding proteins (RBPs), though some have also been implicated in protein–protein interactions and DNA-binding [13]. The significance of further research in this field is underscored by the growing number of publications confirming the pivotal involvement of the CCHC superfamily in RNA metabolism.

Previous studies have categorized *CCHC* genes into nine and seven subfamilies in wheat [21] and *U. virens* [20], respectively. In this study, we screened 450 *CCHC* genes across ten plant species from five different families and constructed a phylogenetic tree, resulting in the division into 17 subfamilies. Interestingly, the distribution of *CCHC* genes among plant subfamilies is uneven. For instance, subfamily VII is exclusive to peaches, while a significant number of genes in subfamilies V and X are predominantly found in *R. rugosa*. Subfamilies IV, VI, VII, VIII, and IX are notably absent in monocot plants, with VI and VII are abundant in Rosa species. This pattern suggests that certain subfamilies may play specialized roles in dicot plants.

Gene duplication is recognized as a pivotal mechanism in plant evolution. The present study demonstrates that DSD plays a pivotal role in the expansion of the *CCHC* gene family in *R. chinensis*. Moreover, the Ka/Ks ratios of all homologous gene pairs in the *RcCCHC* gene family are below 1, suggesting strong purifying selection during their evolutionary history. Gene structure analysis revealed that, unlike other families in roses, such as *WRKY* [29], *AP2/ERF* [30], and *TBL* [31], the *CCHC* gene family exhibits a complex architecture. Most *RcCCHC* genes contain multiple exons, with *RcCCHC23* exhibiting the highest count at 14. Such multi-exon structures often lead to diverse t transcription profiles through alternative splicing, potentially translating into proteins with varied functions, thereby hinting at the intricate roles of the *CCHC* gene family.

Cis-acting elements operate as molecular switches, regulating many biological processes via transcriptional regulation. In our study, we predicted the presence of cis-acting elements in the promoter regions of *RcCCHC* genes, identifying prevalent stress resistance and hormone-responsive elements. Notably, MYB and MeJA emerged as primary signaling molecules associated with plant stress responses and development. Additionally, ABRE and DRE are recognized cis-acting elements that activate gene expression under abiotic stress, with STRE and W-box elements also playing crucial roles in stress regulation. These findings highlight the significant biological role of *RcCCHC* genes in plant stress response mechanisms.

Zinc finger proteins, including those from the *CCHC* gene family, contribute to various functions in plants. For instance, in Arabidopsis, *CCHC* genes are involved in the development of seeds, flowers, and embryos [15]. The RRM structural domain is associated with the *PtoRSZ21* gene, which encodes a serine/arginine-rich splicing factor involved in regulating plant response to drought stress [11]. In addition, the overexpression of *AdRSZ21*, a gene containing the RRM domain in tobacco, induced a hypersensitive response (HR)-like cell death [32]. These pieces of evidence support the role of *CCHC* genes in regulating plant growth, development, and programmed cell death. Notably, genes from subfamily 16, which contain a CSD domain, suggest a role for the *CCHC* gene in plant adaptation to cold stress. For example, genes such as *GhCSP.A1*, *GhCSP.A2*, *GhCSP.A3*, and *GhCSP.A7* in cotton show significant upregulation following low-temperature stress, linking *CCHC* genes to cold resistance [33]. Furthermore, the chickpea genes *Ca04468* and *Ca07571,* homologous to Arabidopsis subfamily 14 genes *AT5G52380* and *AT3G43590*, respectively, are upregulated under drought conditions, further substantiating the involvement of *CCHC* genes in stress response regulation [34]. Overall, these findings collectively highlight the *CCHC* gene family’s involvement in plant growth, hormonal responses, and stress resistance. Nonetheless, the specific molecular mechanisms by which *CCHC* genes regulate these processes in rose remain unclear, necessitating further investigation.

## 4. Materials and Methods

### 4.1. Identification of the Rose CCHC Family Gene

The complete genomic DNA sequence, coding sequences (CDS), proteome sequence, and annotation files for *R. chinensis* Old Blush were sourced from the LIPM browsers Toulouse website (https://lipm-browsers.toulouse.inra.fr/pub/RchiOBHm-V2/, accessed on 26 February 2024). The sources of all other genomic data and annotation files employed in this study are itemized in Appendix A.

To identify the *CCHC* genes in roses, we employed two distinct methodologies. Firstly, the Hidden Markov Model (HMM) profile for the conserved CCHC motif (Pfam ID: PF00098) was retrieved from the Pfam database, and the HMMER3.2.2 software was used to screen the rose whole-genome protein database for matches [35,36]. Secondly, using DIAMOND v2.1.8 software, a BLASP search was performed using the wheat protein sequences of the *CCHC* genes, with a strict E-value threshold of less than 1 × 10^−5^ [37]. High-confidence *CCHC* genes were identified at the intersection of these two approaches. Following this, all redundant sequences and those with an amino acid length shorter than 100 were eliminated [38,39]. Further characterization of these proteins included analysis of molecular weight (MW), instability index, and grand average of hydropathicity (GRAVY) using the ProtParam tool (https://web.expasy.org/protparam/, accessed on 7 March 2024) [40]. Subcellular localization predictions for the *RcCCHC* genes were performed using the Cell-PLoc 2.0 [41]. The nomenclature for the *RcCCHC* genes was established based on their positioning on pseudomolecules, facilitating systemic documentation and referencing.

### 4.2. Phylogenetic Analysis and Gene Structure Analysis

In our study, a total of 450 *CCHC* genes identified from eleven species underwent phylogenetic analysis. Multiple sequence alignment (MSA) of the protein sequences was performed using MUSCLE v3.8.1551. Subsequently, the TrimAI v1.4 was used to refine the alignment. Then, using IQ-TREE v2.2.5, a maximum-likelihood phylogenetic tree was constructed [42,43,44] and the iTOL (https://itol.embl.de/, accessed on 6 August 2024) was used to visualize the phylogenetic tree [45].

Concurrently, using the MEME online program, conserved motifs within the *CCHC* genes were identified, with the analysis revealing 15 conserved motifs (Appendix A) [46]. Using TBtools v2.11.2, the exon–intron structures of the *RcCCHC* genes were displayed, and based on motif distribution and phylogenetic relationships, the *RcCCHC* genes were classified into subfamilies (Appendix A) [47].

### 4.3. Promoter Analysis of RcCCHC-ZFP Genes

To characterize the regions for each *RcCCHC* gene, a 2000 bp upstream fragment was taken from the rose genome and used for promoter analysis. In order to predict the cis-elements, the PlantCARE (https://bioinformatics.psb.ugent.be/webtools/plantcare/html/, accessed on 20 May 2024)was employed [48]. Visualization of these elements was achieved using the Pheatmap package in R v3.4, along with GraphPad Prism v9.5.0 and Adobe Illustrator 2022 for enhanced graphical representation and clarity.

### 4.4. Chromosomal Localization, Synteny, Gene Duplication Analysis, and Ka/Ks Analysis

Chromosomal localization of the *RcCCHC-ZFP* genes was determined from the GFF3 file. Gene duplication events in the rose genome were identified through the DupGen_finder pipeline [49]. Synteny analysis of all CCHC genes was conducted using MSCcanX, with results visualized using the software Circle in TBtools v2.11.2 [50]. The evolutionary relationships among the species were illustrated using the TimeTree online tool (https://timetree.org/, accessed on 27 May 2024). Ka (synonymous substitution rate) and Ks (non-synonymous substitution rate) values for syntenic gene pairs were calculated using TBtools. The duplication time (T) in millions of years ago (Mya) was estimated using the formula T = Ks/2λ × 10^−6^ Mya, where λ = 6.5 × 10^−9^, representing the mutation rate per year per site.

### 4.5. Expression Analysis and Prediction of the Protein Interaction Network of RcCCHC-ZFP Genes

Transcriptome data for leaves and roots of *R. chinensis* under drought stress were retrieved from NCBI (PRJNA663119) [27]. The visualization of expression patterns was conducted tsing the Pheatmap package in R, providing a detailed representation of gene expression changes. Furthermore, using the STRING database, the potential interactions between the *R. chinensis* CCHC protein sequence were investigated (https://cn.string-db.org/, accessed on 20 June 2024), with a medium confidence setting using Cytoscape v3.10.0 software to elucidate the protein interaction network, highlighting key interactions and potential regulatory hubs within the *RcCCHC-ZFP* gene family [51].

### 4.6. Plant Materials and Treatment

The plants of *R. chinensis* Old Blush were sourced from the experimental field of Beijing University of Agriculture. Annual rose branches were cut into small sections, each retaining two buds. These cuttings were treated first with 0.1% carbendazim for 20 min and then with 0.1% rooting powder for 15 min before being planted in a 1:1 potting mix of peat and vermiculite. The cuttings were maintained in controlled conditions at 25 °C with a 16 h light/8 h dark cycle for two months, facilitating optimal root development and growth under consistent environmental conditions.

### 4.7. RcCCHC25 Silencing

The expression of *RcCCHC25* (GenBank ID: PQ114140) was downregulated in rose using VIGS, following the methodology outlined by Jiang et al. [52]. A fragment approximately 350 bp from the 3′ end of *RcCCHC25* was cloned into the pTRV2 to create pTRV2-RcCCHC25 (Appendix A). The Agrobacterium (*Agrobacterium tumefaciens*) strain GV3101 harboring pTRV1 and either pTRV2 or pTRV2-RcCCHC25 was resuspended and mixed in equal proportions. Plant materials were immersed in this suspension under vacuum conditions of 0.7 MPa for 10 min, and it was slowly deflated to 1 atmosphere and repeated twice. Then, it was washed three times with deionized water and incubated in the dark at 8 °C for 3 d. Post-incubation, plants were subjected to 20% PEG6000 solution to simulate drought conditions and were photographed to document phenotypic responses.

Furthermore, we also conducted drought stress experiments using leaves of *Rosa chinensis*. First, we took rose leaves that were growing normally and used the method described above to silence *RcCCHC25*. Post-incubation, these leaves were placed on clean and dry glass Petri dishes for natural drought treatment experiments. Following a 24 h dehydration period and a 3 h rehydration period, leaves were harvested for ion leakage rate analysis to assess cellular damage [4] and the left area was measured using ImageJ v1.54j software.

### 4.8. Quantitative Reverse Transcriptase PCR and Physiological Index Determination

Total RNA was extracted from rose leaves using the RNAprep Kit (TIANGEN, Beijing, China). For cDNA synthesis, the Evo M-MLV RT Mix Kit (Accurate Biology, Beijing, China) was used. Quantitative reverse transcriptase PCR (RT-qPCR) was then performed using the SYBR Green Premix Pro Taq HS qPCR Kit (Accurate Biology, Beijing, China). The RT-qPCR data were subjected to analysis using the 2^−ΔΔCT^ method [53]. In order to restore normalcy to the expression levels, the ubiquitin (*RcUBI2*) gene was employed as an internal control. The reaction system and conditions of RT-qPCR were based on the method proposed by Jiang et al. [27]. All primer sequences used in this study are listed in Appendix A.

## 5. Conclusions

In our study, 41 *CCHC* genes were identified from the *R. chinensis* genome, and their structures, expression patterns, and evolutionary relationships were extensively analyzed. These genes were categorized into 14 subfamilies based on distinct structural motifs and genetic sequences. It has been proposed that DSD is the predominant mode of duplication in the *R. chinensis CCHC* gene family. Gene collinearity analysis showed robust synteny with other dicotyledonous plants, but no significant gene pairs were found with monocotyledonous plants like wheat and microalgae such as *O. lucimarinus*. Additionally, numerous *cis*-acting elements associated with biotic and abiotic stress responses were identified in the promoters of *R. chinensis CCHC* genes. Expression analysis further revealed that *RcCCHC25* was significantly upregulated under drought stress conditions, highlighting its potential role in enhancing drought resistance. The functional role of *RcCCHC25* in rose drought resistance was further corroborated using VIGS. Overall, this study lays the groundwork for further functional characterization of the *CCHC* gene family in rose, particularly its roles in stress responses and plant development.

## Figures and Tables

**Figure 1 ijms-25-08983-f001:**
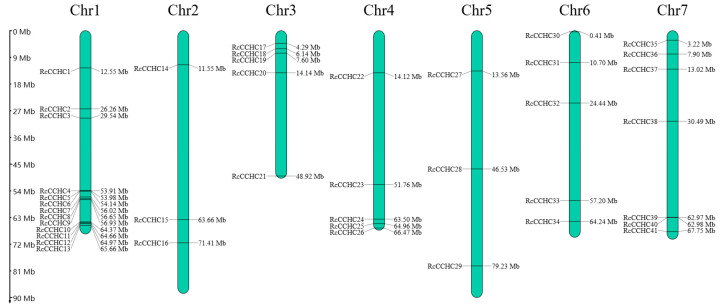
**Chromosomal distribution of *CCHC* gene family in roses.** This diagram illustrates the localization of *CCHC* genes on rose chromosomes, depicted as green rectangles. The chromosome numbers are indicated above each rectangle, while the scale on the left shows the length of such chromosome in megabases (Mb). Positions of *CCHC* genes are precisely marked along the chromosomes.

**Figure 2 ijms-25-08983-f002:**
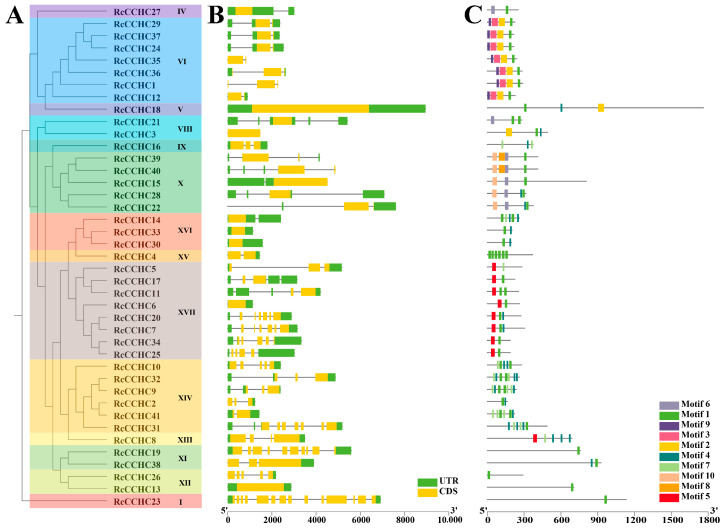
**Phylogenetic analysis, gene structures, and protein-conserved motifs of the rose *CCHC* gene family.** (**A**) Phylogenetic tree of the *CCHC* gene family in rose, with branches color-coded to differentiate subfamilies. Subfamily names are annotated to the right of corresponding gene names. (**B**) Gene structure analysis of the rose *CCHC* gene family. Among them, yellow boxes represent exons, untranslated regions (UTRs) are indicated by green boxes, and introns are indicated by gray lines. (**C**) Distribution of conserved protein motifs within the rose *CCHC* gene family, with each motif depicted by a differently colored box. The length of each line corresponds to the protein length. Motif designations are listed in the lower right-hand corner of the figure.

**Figure 3 ijms-25-08983-f003:**
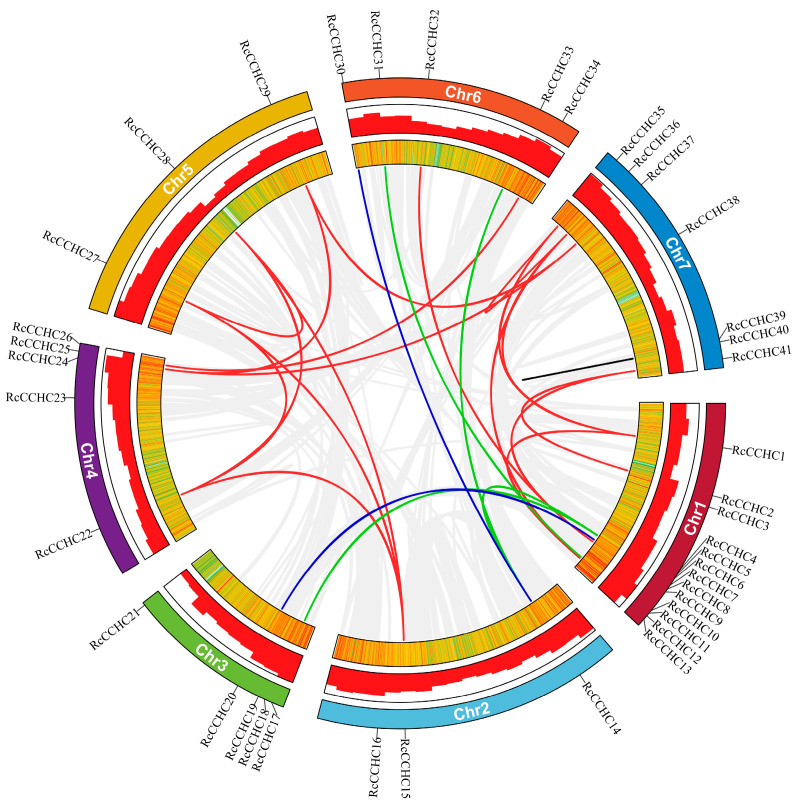
**Gene duplication modes within the rose *CCHC* gene family.** The gray lines represent all homologous gene pairs identified in roses. Lines colored in red, green, blue, and black denote gene pairs resulting from disperse duplication (DSD), retrotransposed duplication (TRD), whole-genome duplication (WGD), and proximal duplication (PD), respectively. The outermost circle represents the seven chromosomes of rose, with the position of *CCHC* genes marked accordingly. The middle circle presents histograms of gene density and the innermost circle features a heatmap indicating gene density across the rose chromosomes.

**Figure 4 ijms-25-08983-f004:**
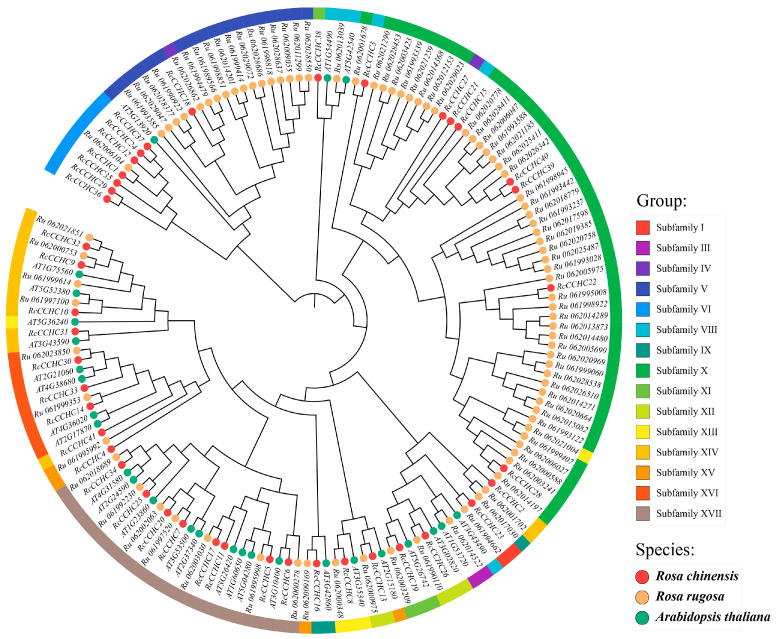
**Phylogenetic tree of *CCHC* genes across three species.** The colors of the outer ring denote distinct subfamilies, and subfamily names are listed on the right. Preceding each gene name, colored circles identify the species, with corresponding scientific names delineated on the right side of the figure.

**Figure 5 ijms-25-08983-f005:**
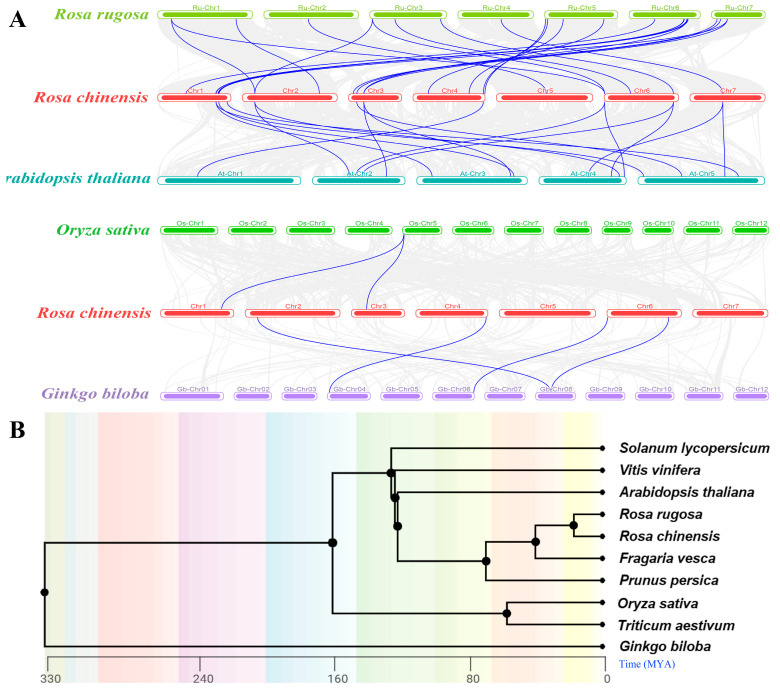
**Collinearity analysis of *CCHC* genes between rose and other representative species.** (**A**) Collinearity of *CCHC* genes between rose and four other representative species. Each rectangle represents a chromosome. Gray lines map all homologous gene pairs between rose and the representative species, highlighting the comprehensive genetic relationships. Blue lines specifically indicate the collinear *CCHC* gene pairs, emphasizing genes with shared evolutionary paths. The names of the species involved are listed on the left side of the panel. (**B**) Evolution tree of rose and nine other species, excluding *Ostreococcus lucimarinus*. The tree visualizes the evolutionary relationships and divergence among the species, based on the analysis of their *CCHC* genes, illustrating both close and distinct genetic affiliations.

**Figure 6 ijms-25-08983-f006:**
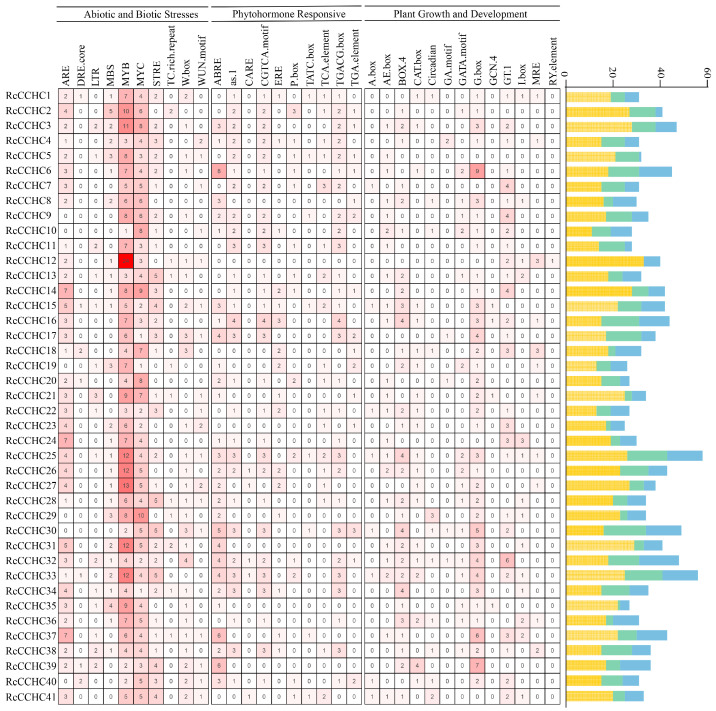
**Prediction of *cis*-acting elements in promoters of rose *CCHC* genes.** The number of each type of element is represented through varying color gradients, offering a visual representation of their prevalence. These cis-acting elements are classified into three functional categories: abiotic and biotic stress, phytohormone responsive, and plant growth and development. The histogram on the right side of the figure uses yellow, green, and blue rectangles to represent the number of cis-acting elements for the three types mentioned above. The length of each rectangle correlates with the quantity of each type of cis-acting element found, providing a quick visual assessment of their distribution and potential regulatory significance in rose *CCHC* genes.

**Figure 7 ijms-25-08983-f007:**
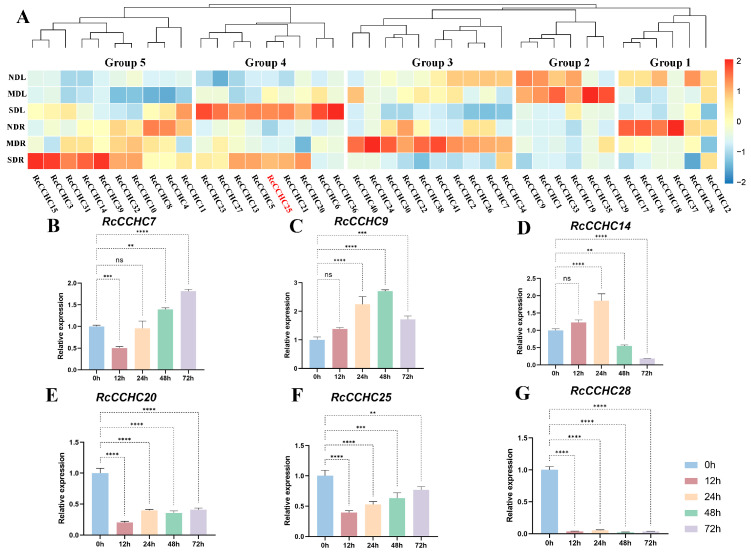
**Expression profile of the *CCHC* gene family in rose under drought stress.** (**A**) ND (no drought), MD (mild drought), and SD (severe drought) indicate the levels of drought stress experienced, while L (leaves) and R (roots) represent the tissue types examined. The genes are organized into five groups according to their expression trends, with each group labeled beneath the dendrogram. (**B**–**G**) Rt-qPCR analysis of six RcCCHC-ZFP family members in rose leaves at five periods of drought. Error bars indicate SE (*n* = 3). Student’s *t*-test was used for all statistical analyses (** *p* <0.01, *** *p* < 0.001, **** *p* <0.0001, ns means not significant).

**Figure 8 ijms-25-08983-f008:**
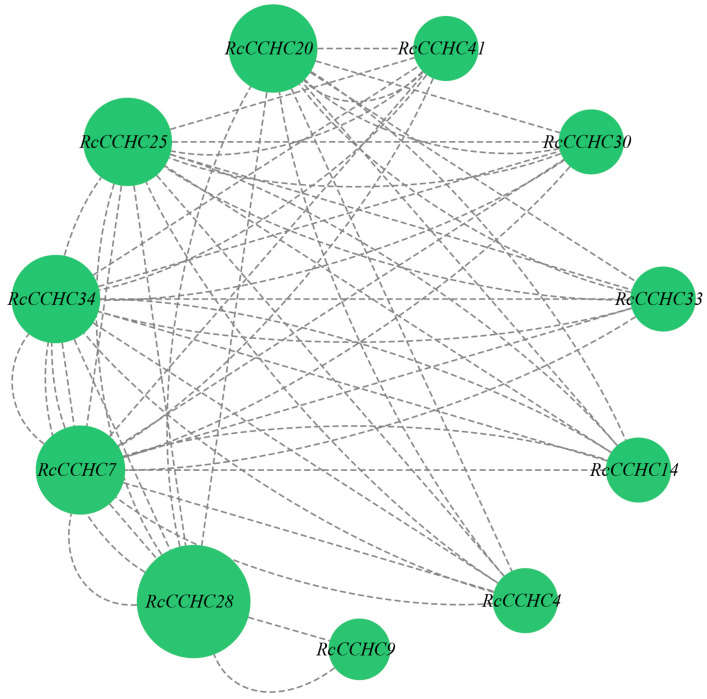
**Protein–protein interaction network of the *CCHC* gene family in rose.** The dashed lines represent interactions between proteins. Each green circle symbolizes a protein, with the circle’s size reflecting the number of interactions that protein has within the network.

**Figure 9 ijms-25-08983-f009:**
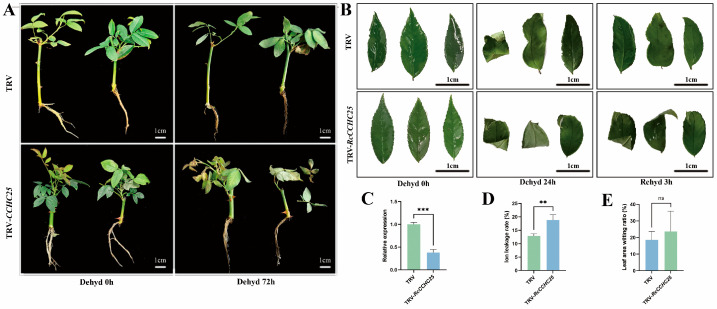
**Functional analysis of *RcCCHC25* gene.** (**A**) Plant phenotypes after dehydration (Dehyd) of TRV (TRV-empty vector) and TRV-RcCCHC25. (**B**) TRV and TRV-RcCCHC25 phenotypes in leaves following dehydration (Dehyd) and subsequent rehydration (Rehyd). Bar = 1 cm. (**C**) comparative expression levels of *RcCCHC25* in both TRV and TRV-RcCCH25 leaves. (**D**) Ion leakage rate. (**E**) Wilting ratio of the leaf area in TRV and TRV-RcCCHC25 leaves. Error bars indicate SE (*n* = 3). Student’s *t*-test was used for all statistical analyses (** *p* < 0.01, *** *p* < 0.001, ns means not significant).

**Table 1 ijms-25-08983-t001:** Molecular features and putative subcellular locations of *Rosa chinensis* CCHCs.

GeneIDs	Chr	Number of Amino Acids	Molecular Weight	Instability Index	GRAVY	Subcellular Iocalization
*RcCCHC1*	Chr1	288	33,405.51	49.85	−0.801	Nucleus.
*RcCCHC2*	Chr1	164	18,609.56	25.94	−0.075	Nucleus.
*RcCCHC3*	Chr1	491	54,875.04	39.5	−0.329	Cell wall. Chloroplast. Nucleus.
*RcCCHC4*	Chr1	370	34,767.92	42.85	−0.927	Nucleus.
*RcCCHC5*	Chr1	284	31,134.43	42.29	−1.287	Chloroplast. Nucleus.
*RcCCHC6*	Chr1	263	29,012.75	46.46	−1.21	Chloroplast. Nucleus.
*RcCCHC7*	Chr1	306	34,712.85	108.65	−1.666	Chloroplast. Nucleus.
*RcCCHC8*	Chr1	691	74,605.05	47.12	−0.693	Chloroplast. Nucleus.
*RcCCHC9*	Chr1	237	25,939.47	70.38	−0.659	Nucleus.
*RcCCHC10*	Chr1	281	31,032.34	38.97	−1.057	Golgi apparatus. Nucleus.
*RcCCHC11*	Chr1	255	27,528.53	36.58	0.012	Chloroplast.
*RcCCHC12*	Chr1	228	26,211.52	39.59	−0.811	Chloroplast.
*RcCCHC13*	Chr1	705	80,841.2	54.63	−0.736	Nucleus.
*RcCCHC14*	Chr2	262	25,698.68	36.79	−0.636	Golgi apparatus.
*RcCCHC15*	Chr2	808	90,820.67	43.94	−0.35	Chloroplast.
*RcCCHC16*	Chr2	374	38,166.98	42.92	−0.513	Cell wall.
*RcCCHC17*	Chr3	225	23,856.57	46.12	−1.248	Chloroplast. Nucleus.
*RcCCHC18*	Chr3	1762	200,188.9	36.35	−0.56	Nucleus.
*RcCCHC19*	Chr3	759	82,244.5	45.9	−0.353	Mitochondrion. Nucleus.
*RcCCHC20*	Chr3	276	31,962.26	102.42	−1.524	Chloroplast. Nucleus.
*RcCCHC21*	Chr3	283	31,815.36	56.83	−0.642	Nucleus.
*RcCCHC22*	Chr4	375	42,256.56	49.14	−0.34	Chloroplast.
*RcCCHC23*	Chr4	1133	126,455.36	43.01	−0.662	Mitochondrion. Nucleus.
*RcCCHC24*	Chr4	217	24,829.79	27.13	−0.638	Chloroplast.
*RcCCHC25*	Chr4	188	21,171.57	95.71	−1.22	Chloroplast. Nucleus.
*RcCCHC26*	Chr4	291	32,523.14	46.83	−0.991	Nucleus.
*RcCCHC27*	Chr5	253	29,110.08	41.92	−0.821	Nucleus.
*RcCCHC28*	Chr5	316	35,659.93	39.62	−0.344	Nucleus.
*RcCCHC29*	Chr5	226	25,836.14	28.69	−0.659	Nucleus.
*RcCCHC30*	Chr6	199	18,907.95	42.17	−0.772	Nucleus.
*RcCCHC31*	Chr6	488	54,240.26	46.88	−0.845	Nucleus.
*RcCCHC32*	Chr6	262	28,691.48	72.43	−0.818	Golgi apparatus. Nucleus.
*RcCCHC33*	Chr6	202	20,575.29	52.35	−0.826	Nucleus.
*RcCCHC34*	Chr6	188	20,953.06	104.34	−1.326	Chloroplast. Nucleus.
*RcCCHC35*	Chr7	241	27,798.22	32.46	−0.748	Nucleus.
*RcCCHC36*	Chr7	284	32,523.23	39.82	−0.766	Chloroplast. Nucleus.
*RcCCHC37*	Chr7	217	24,739.58	30.19	−0.708	Chloroplast. Nucleus.
*RcCCHC38*	Chr7	927	101,603.41	41.97	−0.447	Nucleus.
*RcCCHC39*	Chr7	413	45,598.95	49.1	−0.819	Nucleus.
*RcCCHC40*	Chr7	413	45,598.95	49.1	−0.819	Nucleus.
*RcCCHC41*	Chr7	223	24,159.34	50.59	−0.784	Nucleus.

## Data Availability

All data, tables, and figures in this manuscript are original, and are contained within the article and Appendix A.

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
