# Peer review of "Genome-Wide Analyses of CCHC Family Genes and Their Expression Profiles under Drought Stress in Rose (Rosa chinensis)"

_ijms, 2024, doi:10.3390/ijms25168983_

Round 1

Reviewer 1 Report

Comments and Suggestions for Authors

 In the manuscript named “Genome-Wide Analyses of CCHC Family Genes and Their Expression Profiles under Drought Stress in Rose (Rosa chinensis)”, Shijie Li et al have performed genome-wide analysis of CCHC genes in rose, including bioinformatics analysis, qRT-PCR, and gene silencing to validate one member, RcCCHC25, its function in response to drought stress. Their results were meaningful for rose genetic breeding works, and their findings were also helpful for plant genetic improvement in future. But there were some comments about their research.

Major,

(1) Authors have performed many bioinformatics analysis works, but little wet experiments in this manuscript. Even some bioinformatics is redundancy, for example, Theoretical pI, etc.

(2) Authors have selected RcCCHC25 for validating its function, the deduction was not insufficiency, just based with more cis-active elements. But the PlantCARE results was not precise. In addition, authors have analyzed RNA-seq data, but there were some members were both response to drought stress in leaf and root, why didn’t selected.

(3) Authors have described as RcCCHC34 (line 294 to 297), please check them, and supply all sequencing results and primers in supplements.

Minor,

(4) In table 1, many gene information was missing, such as gene ID, chromosomes.

(5) In genome-duplication analysis, authors have characterized gene duplication as disperse duplication (DSD), retro-transposed duplication (TRD), whole-genome duplication (WGD), and proximal duplication (PD), did authors have checked these results, which is TRD, please display the results in detailly.

(6) In figure 4, the nodes were too small to displaying, please remove this figure into supplements. Meanwhile, reconstructed the tree using two or three species.

(7) I can’t get meaning of figure 5B, is it consisted with plant evolution process? or just a plot?

(8) Authors have analyzed RNA-seq data, but they haven’t validated these genes with qRT-PCR. In addition, most CCHC genes have different expressional profiles in response to mild drought and severe drought, it was unusual, please check them expressions.

(3) “Rehyd” in figure 9, please check it.

Reviewer 2 Report

Comments and Suggestions for Authors

The rose is an important ornamental flower of economic importance. Due to climate change, abiotic stressors, especially drought events, are increasing worldwide, and screening tolerant genes against drought could be useful in this regard. In this study, the authors thoroughly studied the CCHS gene family in roses under drought stress and presented the data in a well-mannered way. The paper could be accepted after this minor revision.

Comment 1: Line 12: Abstract: After the first line, please add one line about the rose and its importance.

Comment 2: Line 26: Keywords: Change the keyword to: zinc finger proteins, cut flowers, abiotic stress, VIGS, gene family analysis.

Comment 3: Lines 63–68: Introduction: Please move these lines to the beginning of the introduction and combine these lines to show how drought stress impacts rose species at the physiological, biochemical, and molecular levels.

Comment 4: Line 99: Figure 1: Change the Chromosome to Chromosomal. Moreover, please make the width of each chromosome a bit larger and also make the chromosome numbers bold, so it would be better for reading.

Comment 5: I don't know why, but supplementary figures are not provided with the file; please also provide supplemental data in revision so the reviewers can also analyze that.

Comment 6: Line 182: Figure 4: The phylogenetic tree is very complex, and even the names are not properly readable, so I would suggest moving this phylogeny into the supplementary data and constructing a new phylogenetic tree that contains fewer plant species, e.g., Rosa Chinensis, Rosa Rugosa, and Arabidopsis.

Comment 7: Figure 7: As RcCCH25 is the gene of interest, please highlight this in the figure for better reading. Moreover, please explain in detail in the results why RcCCH25 was selected for functional study, as there are many other genes as well that have good expression.

Comment 8: Figure 9: Instead of providing pictures of only leaves, please provide the full plant pictures.

Comment 9: Lines 420–430: Provide details on the methodology by which the gene duplication was performed and, secondly, how the evolutionary relationship tree was generated.

Comment 10: Lines 441-447: Please provide details on how the drought stress was applied to rose plants.

Comment 11: In the supplementary files, do provide the proteins and gene sequences for the plant species studied.

Comment 12: The plagiarism is too high; please decrease the plagiarism to below 20 percent.

Reviewer 3 Report

Comments and Suggestions for Authors

The manuscript is very interesting in its focus, but it is very problematic to perform an adequate review of the manuscript. In particular, evaluate the results and their discussion. Why? The authors refer to Supplementary Materials in the text, but these are not available. That's why I evaluate these two parts (Results/Discussion) more in terms of form than content.

Keywords - some match the title, which is good to remove.

Introduction - in my opinion, it is short and does not cover the whole issue, e.g. (line 40-62) they describe the meaning, use and interaction of ZFPs is very general. E.g. transcriptomic studies of barley vs. drought stress (DOI: 10.17221/69/2022-CJGPB) on lines 50-52. This paragraph should also include a description of the importance of phytohormones, as the authors discuss this in their study. E.g. recently, a lot of attention is paid to strigolactones in plants and resistance to drought, salinity, etc. (DOI: 10.17221/88/2023-CJGPB).

Material and Methods - is also incomplete, e.g. in sections 4.1. until 4.3. references to the given procedures are missing (only lines are documented). Line 443 - "carbendazim" treatment (concentration) etc. is not precisely specified. 4.8. RT-qPCR lacks reaction conditions or references to the procedure used.

The Results and Discussion section should only be evaluated after the possible delivery of Supplementary Materials. On the formal side, it is advisable to follow the standards in writing, e.g. Arabidopsis - in italics, genes in the text, tables (Table 1) and figures (e.g. 8), etc. Figure 9A lacks a scale, B-D - lacks a description of what the line represents. Variability is characterized by what?

Based on the above facts, I cannot recommend the manuscript for publication in its current form. It must be delivered in complete form!

Round 2

Reviewer 1 Report

Comments and Suggestions for Authors

Thanks for authors’ work, most of my comments were systematically addressed, I have no more comments about it. Good luck.

Author Response

Thank you very much for your positive comments.

Reviewer 3 Report

Comments and Suggestions for Authors

The authors accepted all my comments. Regarding the current form, I have comments/remarks on the formal form, where a large number of small adjustments need to be made:

Line 12 - Rosa (italics).

Line 41-42 - transcription factors - are the authors describing proteins or a region of the genome? If a region of the genome, it should be in italics.

Line 44 - ABA - first use and the abbreviation in brackets must be explained.

Line 72 - C2H2 - again genome region or proteins (format resolution, i.e. genome region in italics).

Figure 1 - the format has changed, i.e. black background and you cannot see what the authors describe in the legend and was part of Fig. 1 in the first version of the manuscript.

References - No. 18 (line 587) - the name of the journal is in full form, but the others have a shortened format (must be edited according to the instruction for authors).

Figure S1 - should be the main title, as is the case with the others and only then split into A and B.

Table S1 - chromosome number (space missing); common name - maybe better to use species; Japonica - is properly japonica; annotated RabGAP in the legend is missing.

Table S2 - missing explanation in CDS and UTR legend.

Table S3, S4 and S6 - Japonica - properly japonica.

Table S4 - missing explanation for NaN or /, which are used in the table.

Table S4 - explanations for WGD, PD, TRD and DSD are missing.

Round 3

Reviewer 3 Report

Comments and Suggestions for Authors

The authors corrected all my comments.

Perhaps the last thing that should be fixed:

Table S4 - authors have "Na" in the legend, but the value is "NaN" in the table.

This is a small thing that can be corrected as part of the author's proofreading, and therefore I can recommend the manuscript for publication.